# A Feedback Analysis Automation Using Business Intelligence Technology in Companies Organizing Urban Public Transport

Michał Bańka [1], Jakub Daniłowski [1], Mirosław Czerliński [2], Jakub Murawski [2,*], Renata Żochowska [3] and Aleksander Sobota [3]

1　Faculty of Mechanical and Industrial Engineering, Warsaw University of Technology, Narbutta 85 Str., 02-524 Warsaw, Poland
2　Faculty of Transport, Warsaw University of Technology, Koszykowa 75 Str., 00-662 Warsaw, Poland
3　Faculty of Transport and Aviation Engineering, Silesian University of Technology, Krasińskiego 8 Str., 40-019 Katowice, Poland
*　Correspondence: jakub.murawski@pw.edu.pl

**Abstract:** In companies organizing urban public transport, passengers' feedback is usually collected personally, by phone, by e-mail, or via the Internet. Business Intelligence tools enable this process by automating the data flow between systems used to collect, process, and analyse data by applying analytical Business Intelligence tools. The most important advantage resulting from implementing the tool is facilitating contact between the company and passengers, which allows for an immediate response to the information reported by passengers by introducing the changes in passenger service offered by the company. Another advantage of the discussed tools is the ability to analyse the collected data quickly and easily. Due to the low implementation costs, the developed tool is dedicated mainly to the companies organizing urban public transport. The article presents an example of the use of Power BI tools in the Metropolitan Transport Authority, a company that is the largest organizer of public transport in Poland.

**Keywords:** business intelligence; urban public transport; data analysis; Power Automate; Power BI; passenger transport

## 1. Introduction

The modern world is based on countless data streams of various kinds. They should be collected, structured, analysed, and visualized to use them effectively. Only accomplishing all the steps mentioned above allows correct conclusions to be drawn. Collected data may relate to many aspects of a company, such as the financial situation, sales volume, customer satisfaction, market position, or process efficiency. Data analysis and visualization substantially support decision-making processes, which are key for company development. Business Intelligence (BI) is an answer to the needs regarding using available data, especially in the field of customer service [1]. The steadily growing interest in BI tools has been noticeable worldwide for many years. The technology has a low entry barrier creating space and opportunity for companies that want to improve processes within their organizational structures significantly. The idea of smart cities makes it possible to use available data to provide better transport services. What is essential is that the data generated every day by public transport can be used to gain a better insight into improving the quality of services offered. For instance, their application can be used to create new routes, reorganize existing transport lines, or collect information on passenger behaviour [2].

The Business Intelligence project for the public transport system is undoubtedly a challenge. The BI-based transport system would generate millions of registers per month, monitoring users every min [3].

There are several internationally recognized transportation companies that use BI to support their services. For example, Sybase, Oracle, and Opalis are solutions provided by

companies offering their services to the public transport industry. Fujitsu Consulting has implemented a solution for South African Airways in South Africa. Sybase for American Airlines in the US and Oracle Consulting Services were responsible for implementing a BI solution for China Eastern Airlines. These companies initially had problems with a large volume of information affecting their business activities, thus hindering the decision-making process. Only the implementation of BI systems improved the situation [4].

Information received from customers is extremely important from the point of view of any company. In the case of entities dealing with the urban public transport organization, feedback allows us to measure the level of passenger satisfaction and collect their comments and observations about the services offered. These data determine the possibility of defining the appropriate directions of the company's development and improving passenger service by offering them forms of services adjusted to their needs and expectations. Therefore, the constant collection of feedback, its analysis, and formulation of correct conclusions are essential in increasing the level of passenger service and encouraging them to use public transport, which is a vital element of a sustainable transport policy in many administrative units, including countries, provinces, regions, or cities [5–7].

In the case of many companies organizing urban public transport, collecting feedback from passengers takes place personally during dedicated meetings, by phone, by e-mail, or via the Internet. Therefore, the paper aims to present a BI solution that can be an effective tool for automatically collecting passengers' opinions or experiences on the services offered. BI tools allow feedback collecting and analysing processes to be automated, enabling decision-makers to better understand the expectations of the passengers. The collected information can be used as support in making decisions regarding the improvement of the offered services in terms of meeting the principles of sustainable development. The proposed solution assumes using Power Automate and Power BI applications, which, by combining analytical and automation functions, make it possible to optimize company resources' use and reduce human errors in the data processing.

The article is divided into six chapters. Section 2 provides a short description of BI technology and describes its most important functionalities. Section 3 provides an overview of conventional non-BI solutions currently used to collect customer feedback. Section 4 focuses on an overview of BI technologies and tools. Section 5 presents the model of tool based on BI technology, while Section 6 describes an example of using BI technology in the Metropolitan Transport Authority, a company that is the largest organizer of public transport in Poland. The discussion and conclusions are presented accordingly in Sections 7 and 8.

## 2. Business Intelligence Technologies

BI is not a new issue. The first information in this area has already appeared in the publication of Luhn in 1958 [8]. BI class systems have evolved from IT Management Systems, such as management information systems or decision support systems. BI combines areas and technologies such as statistics, econometrics, operational research, artificial intelligence, databases, business reporting, analytics, data mining, and benchmarking.

Subsequent publications indicated that the intelligence of an organization should be understood as collecting data and processing information contained therein for the needs of a given organization [9]. Eventually, in 1989, H. Dresner (Gartner Group) defined BI as a set of concepts and methods to support decision making in a company using knowledge-based systems [10]. It can be said that BI eliminates all the complexity associated with business processes and converts it into data models invisible to an ordinary recipient of a report [11]. Thanks to BI technology, it is also possible to automate data analysis, which allows for data modelling, forecasting, and benchmarking that take into account a large number of factors influencing the company's situation, such as market competitors, customer satisfaction, quality of provided services, or environmental impact [12,13].

BI solutions support generating knowledge that is useful in building the company's position. In a narrower sense, these are databases, applications, and systems whose goal is

to create valuable analyses that help increase the efficiency of the enterprise. BI technologies include various systems. The most popular are information systems in management, decision support systems, and geographic information systems.

The main element of BI are performance indicators, often called Key Performance Indicators (KPIs). When developing a strategy for controlling the areas of its activity, a company should define financial and non-financial indicators. They are an essential element of monitoring processes in the organization because they allow us to decide whether the assumed goals are being achieved and at what stage they are.

It is worth noting that data analysed by BI tools have been available to companies for a long time. Still, only an appropriate association of data sets and presentation of interrelations between them in an easy-to-understand manner allows new conclusions to be drawn. Therefore, thanks to the presentation of available data in a unique, more accessible, and readable form, BI facilitates an understanding of the company's business processes and environment [14]. Thus, BI tools can be a source of a company's competitive advantage.

There are three basic processes in BI tools:

- Data extraction;
- Data transformation;
- Data import.

The first of the described processes consists of the qualitative and quantitative identification of data necessary to perform a given analysis and then identification and selection of data sources. The second stage is data transformation, which consists of verifying the analysed data set by clarifying any inaccuracies, removing redundant data, and validating data correctness. In the third stage, validated and structured data are imported to the database, on which analyses will be performed using the BI tool. The duration of individual stages and the consideration of any additional sub-stages depends on the size of the data set and the purpose of the analysis [15]. Due to its complexity and many issues that should be considered, creating a database requires a system approach. Creating a database should be preceded by determining a set of dependencies characterizing individual data structures [16]. In addition, it is worth paying attention to the necessity of continuous database development, which extends the scope of conducted research and analyses [17]. A previously defined data management system should be responsible for processing data in a database [18].

BI has become especially important in recent years due to a growing number of databases and potential data sources resulting from digitization covering all aspects of life. In addition, the latest BI tools offer new opportunities and previously undiscovered functionalities that do not require expert IT knowledge. Therefore, BI is no longer the domain of large international corporations but is more commonly used by other companies to better understand business processes and changes in the environment [19].

The advantages of using BI for companies dealing with the organization of urban public transport are similar to those for large corporations. The most important of them include:

- Support for making strategic decisions (e.g., regarding changes in the transport offer);
- Optimization of business processes through efficiency analysis;
- Optimization of a company's resource utilization by minimizing the labour intensity of the analyses performed;
- Acquiring additional knowledge about the market and financial situation of the company and its socio-economic environment [20].

Organizing urban public transport involves the generation of large sets of data related to fleet management, route and timetable planning, and the assessment of the use of the transport services' use level. The information obtained should be analysed in detail. This is a necessary action that allows the company's operations to be optimized, thus increasing the level of matching the offer to customer expectations. Performing such analyses based on dispersed data obtained from various sources and compiled in various forms is often inconvenient and time-consuming, and the results can be unreliable. Therefore, it is worth

using the tools supporting the automation and improvement of the analytical process, which have been successfully used for many decades in large corporations.

### 3. A Review of Conventional Methods Used to Collect Customers' Feedback

As mentioned in the introduction, the companies that do not use BI tools usually collect feedback from the customers in the following ways:

- By e-mail correspondence;
- Organizing meetings, videoconferences, or telephone calls;
- By single-use forms;
- Using the Internet.

#### 3.1. An E-Mail Correspondence

One of the mechanisms of collecting feedback is communication with the customer via e-mail correspondence. It assumes that customers will send comments and observations regarding the product or service to the e-mail address indicated on a company's website or in documentation accompanying the product (e.g., user manual). However, this method does not allow customer data to be anonymized, as the sender has to disclose at least his e-mail address. Consequently, customers may have concerns about articulating their comments if they cannot do so anonymously [21]. The factor influencing the low effectiveness of the described method is also the fact that in the case of large corporations, the number of e-mails received is so substantial that it becomes problematic to process them all, analyse and draw correct conclusions.

#### 3.2. The Meetings, Videoconferences, or Telephone Calls

Another method of collecting customer feedback is direct contact, a telephone conversation, or a remote connection via video chat or videoconference. In the case of direct communication, the most common practice is to send an invitation to a dedicated meeting where the client can talk to the company's representatives and express his opinion. The advantage of this method is a possibility of a detailed explanation of all issues and a chance to ask clarifying questions by the company's representative [22].

However, the described method also has its limitations. The most important ones are:

- It is time-consuming.
- The necessity to provide a convenient place to organize a meeting (does not apply to telephone calls and videoconferences).
- The necessity to prepare a note from a meeting or conversation to pass obtained information to decision-makers.

Therefore, the described methods are most often applied to small projects involving a small number of customers. If a company's activity is mass-scale, organizing meetings or telephone conversations with many customers is not an effective method of collecting feedback.

A modern form of obtaining feedback from customers is the use of bots that conduct conversations. Automatic phone call systems increase the efficiency of a process by minimizing the number of people serving customers [23]. Intelligent systems are taught essential solutions to problems and how to collect customers' feedback. When the bot cannot solve the problem, the client is switched to a consultant who reacts only in emergencies [24].

#### 3.3. Single-Use Forms

Another method of collecting customer feedback is a dedicated single-use form. This is a partially automated method which generates measurable data. Communication with a customer via a single-use form is similar to contact through e-mail. Single-use forms allow questions and templates of possible answers to be defined. Single-use forms can be prepared in any application with such a feature (e.g., Microsoft Forms). Unfortunately, single-use forms are not fully automated, as it is usually necessary to send customers

an invitation with a link to the form. This method is the closest to the BI tool presented in the paper.

### 3.4. The Internet

The Internet is becoming a significant form of obtaining feedback from mass customers. A considerable part of society has access to it. People spend more time in this space every year, so companies also use it for business activities. There are more and more channels for collecting feedback and the development of web portals and mobile applications. Primary channels for collecting customers' feedback are:

- Customer opinions and ratings on the Internet (on portals such as Trip Advisor, Google Maps, platforms for selling services such as Booking.com or Booksy);
- Chatbots on social media;
- Comments and reactions to company posts on social media;
- Marketing campaigns focused on a client's activities (e.g., publication of photos, videos, relations with hashtags on social media, etc.);
- Articles or press releases (especially when offering services of general interest).

In the sector of services assigned to specific places of service provision, portals and applications allow users to evaluate and comment on the quality of services offered. Customers assess the quality of service on a multi-level scale and can publish any comments and share experiences related to a service provided. Some of these platforms are open (such as Google Maps and Trip Advisor [25]), so each user can evaluate the place, whether he has visited it or not. Some others require a service (e.g., Booking.com or Booksy), which can only be assessed after completion [26]. Opinions can be extremely valuable for service providers to improve and match them better with customer needs. However, they may also carry false information that is difficult to verify. On the other hand, ratings are given by the users' position companies, so with a sufficiently large number of ratings issued, you can be sure of the quality of offered service [27].

Another tool used by companies is intelligent chatbots for customer service. A chatbot has programmed conversation paths with the client and uses machine learning algorithms to improve its response [28]. A tool also collects feedback on an offered service. Data on customer responses are saved in a database that can be further analysed. Significantly, a solution reduces the need to use real people to conduct a conversation. However, in unusual situations, a chatbot may not be able to complete a conversation due to a nature of a problem [29].

Companies use various social media (e.g., Facebook, Instagram, TikTok) to provide information about services or products offered and buy advertisements to reach potential users. Comments under such promotional campaigns also provide essential feedback on the feelings and needs of customers regarding offered products [30].

Another form of promoting a company and reaching feedback may be the involvement of users on social media to promote a product. For example, these may be competition campaigns encouraging the publication of materials related to a service or product in return for specific benefits. In this way, on the one hand, a company learns what users value in offered products or services (through posts and photos on Facebook, videos, and reports on Instagram or TikTok). On the other hand, it is a form of reaching new customers [31].

In the case of general interest services, a company's activities may be interesting from the perspective of media and journalists. A form of feedback is articles or video materials presenting or even evaluating new solutions. In this case, feedback is processed by a specialist creating materials or articles, which sometimes also reaches individual users. Journalists can strongly influence the public, so it is also an essential source of feedback.

### 3.5. A Summary of Conventional Methods Used to Collect Customers' Feedback

Presented traditional mechanisms of collection feedback from a client are simple, widespread methods and do not require sophisticated IT architecture. However, they have many disadvantages. The following disadvantages should be mentioned:

- Limited possibilities of handling large data volumes;
- No possibility of automated data structuring;
- Difficulties in analysing obtained data;
- Time-consuming preparation of data visualization;
- Susceptibility to human errors.

Table 1 compares specific methods used to collect customers' feedback.

**Table 1.** Comparison of different methods used to collect customers' feedback.

| Functions/Features | E-Mail Correspondence | Meetings, Video Calls | Single-Use Forms | Internet (Social Media) | BI-Based Solution |
|---|---|---|---|---|---|
| High quality and completeness of collected data. | X | X | X | | X |
| Repeatability and automation of the data collection process. | | | | | X |
| Low time-consuming data collection and processing. | | | X | X | X |
| Ease of collected data analysis and visualization. | | | | X | X |
| An option of refreshing the collected data (process continuity). | | | | | X |

Therefore, the presented methods are used primarily in companies that do not cooperate with many customers and do not have specific development plans. In entities that, for example, plan to scale their operations, enter new markets or significantly increase the number of customers served, the willingness to serve all new customers using the mechanisms mentioned above will be associated with a significant increase in the costs of collecting feedback from customers and a need to hire new people dedicated to the implementation of tasks related to it. In addition, they are often costs that cannot be planned or included in a budget.

A communication form with a client also depends on the nature of the company and the service provided. The basic breakdown consists of services provided directly to consumers, services for producers and businesses, and services of general interest. Consumer services include trade, repair, hotels, restaurants, passenger transport, communications, personal services, utilities, education, health, social services, recreation, culture, and sport. Services for producers and businesses include transportation of goods, warehouse management, financial brokerage, real estate services, rental of machinery and equipment, IT and related activities, research and development, and other business-related activities. In contrast, services of general interest include public administration, national defence, activities of member organizations, national organizations, and teams [32]. In each of these areas, an emphasis on different tools used differs.

In the case of companies dealing with the organization of urban public transport, planning to expand their activities and improve the quality of transport services, it seems reasonable to introduce modern BI solutions. Such tools would enable the automation of obtaining feedback from passengers and improve data processing and analysis, providing decision-makers with helpful information to support decisions on further actions to meet the needs of current and potential passengers. An additional aspect favouring BI technology is the ability to efficiently adapt to all kinds of changes (e.g., unexpected market changes).

## 4. A Review of Business Intelligence Tools and Technologies

Currently, many data science and business intelligence organizations are considering entering the world of BI platforms. Figure 1 shows the "Magic Quadrant for Analytics

and Business Intelligence Platforms" from the 2022 annual report of Gartner Inc., which presents the current situation in the BI tools market.

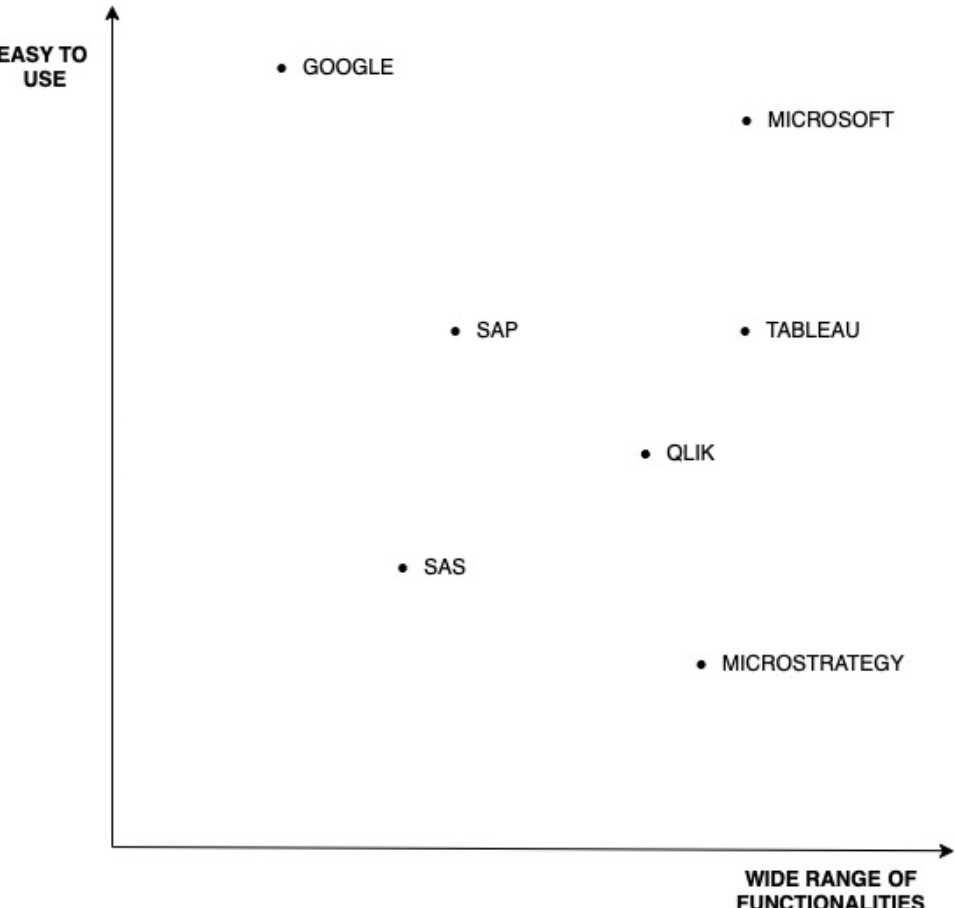

**Figure 1.** Comparison of Business Intelligence platforms.

According to the report, there are four categories of technology providers:

- "Leaders" are companies that have the largest market share and dominate in terms of functionality offered and the possibility of integrating their tools with other providers of interdependent services.
- "Visioners" are entities that can be considered as leaders of innovation in terms of new functionality and usability for users.
- "Niche players" are organizations that create BI software that is not used extensively (their tools are highly specialized and problem-oriented).
- "Challengers" are companies whose BI platforms aspire to be leaders regarding the number of users and offered functionalities.

Technologies and Power BI platforms offered by Microsoft, as well as Tableau (Tableau Desktop version 2021.1.1 by Tableau Software Inc., Seattle, WA, USA) and Qlik (Qlik Sense version 2.1 by Qlik Technologies Inc., King of Prussia, PA, USA) software, are currently listed as market leaders. All of the mentioned companies offer comprehensive platforms for data visualization analysis. Additionally, the Power BI platform combines several tools in one service. In the case of many other solutions, several components must be installed and used separately, which can be an essential factor in determining a company's choice of a specific BI technology [33,34].

Markets and Markets Research estimates that the BI services market will reach USD 33.5 billion by 2025 [35]. This estimation gives broad prospects for mature enterprises with an established position that will introduce BI solutions to their operations and for start-ups that will provide support services based on BI technologies [36]. In the future, BI

tools will be even more automated, more widespread, error-limited, and, above all, more user-friendly. One of the factors that may have the most significant impact on market development is a connection of BI technology with the collection, processing, and analysis of Big Data, i.e., an analysis of much larger data sets (terabytes or petabytes, depending on the industry) [37].

### 4.1. Characteristics of Automation Tools

Automation is the ability to automatically perform repetitive processes or tasks, which allows us to focus on functions of the most significant importance. For example, an application running in the cloud can be treated as a way to unify many different services in a cloud, which leads to the structuring of processes and centralization of functions in a company. The main advantages of business process automation are:

- Increased efficiency—automation significantly improves efficiency, which allows the same job to be done in less time or more tasks to be completed at the same time.
- Increasing accuracy—automation enables the elimination of human errors related to data processing, but it requires additional testing of developed tools, additional training, and tool supervision by qualified employees.
- Increased security—access to data is well controlled and documented, and the risk of data leakage is much lower than in the case of tasks performed by people.

There are several popular business process automation tools on the market. Unfortunately, many of them are only a tiny part of the existing infrastructure of a given provider, so they are not always open to regular improvements or bug fixes. Therefore, it is worth analysing whether the project implemented in a company will be based on one leading technology or composed of many unrelated components. A planned project may be more open to suppliers offering automation tools in the second case.

Among the applications concentrated on the IT industry and focused on subsequent reporting of automated processes, for the solution described in the paper, a choice was made from two tools: Power Automate and Nintex. A selection of Power Automate was made due to consistency and integrity with the tools offered by Microsoft (Office 365, Power BI, and Microsoft Forms). In addition, thanks to better-debugging support (in this case, understood as a process of systemic reduction of several errors in software) and a faster flow testing option, it was decided that Power Automate would be a safer solution than the tool provided by Nintex.

### 4.2. Characteristics of Survey Tools

As mentioned before, there are many tools for creating surveys and forms, such as Google Forms and Microsoft Forms for business use. Notably, in most market survey tools, subscription fees are negligible compared to potential profits from using these tools. Therefore, a key criterion for selecting a survey tool is the ability to synchronize with other applications or systems used in the target system. It is a factor that determines efficient data transfer and its subsequent analysis.

The first discussed tool is Google Forms, which offers one of the fastest and most intuitive ways to create forms. The application makes it possible to choose from several fields that can be used in a form or a survey tool. In addition, Google Forms has simple data validation functionality for text fields and conditional logic to handle different questions based on previous answers [38].

Microsoft Forms is, like Google Forms, an intuitive, very advanced, and free application. Its operation is based mainly on the same principles as Google Forms. Still, it has more possibilities for data analysis thanks to integration with Microsoft Excel, which may be crucial in working with other Microsoft tools used in enterprises. The number of ways to analyse survey results distinguishes Microsoft Forms from other survey tools. Microsoft Forms offers much more of them. In addition, in Microsoft Forms, there is a quick overview of answers and in-depth analysis in an Excel spreadsheet (or in Power BI). Power BI makes it possible to obtain a more advanced analysis and visualization of the received data [39].

Considering the arguments mentioned above, the best tailored to the characteristics of a feedback analysis problem is the Microsoft Forms tool, which provides the most intuitive integration with Microsoft Excel spreadsheets [40].

## 5. The Model of Tool Based on Business Intelligence Technology

### 5.1. General Concept

Conducted literature analysis has shown that it is possible to implement advanced IT systems based on automation and Microsoft BI tools [41]. Using this type of software, it is possible to implement many side tools that will act as one comprehensive application for company feedback analysis. An implementation of a discussed solution may bring the following benefits:

- Information from customers will be collected automatically, so time that was previously needed to obtain data (preparing an e-mail, organizing a meeting, or other types of communication) will be reduced to zero.
- Analyses will be carried out automatically. Thus, there will be no need to develop data and analyse them using ineffective spreadsheets manually.
- A possibility of convenient verification in a structured repository will be ensured in case of necessary data control or verification.
- Data in a report will be refreshed automatically; thus, the user will be able to rely on the most recent data from customers at any time.

The proposed model solution assumes that the tool will be fully automated and easy to maintain so that it will not generate additional maintenance costs. It is also necessary to create an infrastructure that will be auto-scalable regardless of a data stream. The presented solution will ensure a dynamic allocation of resources to match demand.

In addition, the tool will have a good-looking visual side on both sides: company and customer. The layout of the forms will be tailored for clients, and the report will be customized for the company's requirements. Due to consistency between front-end and back-end sites, a company will have a complete and professional tool for everyday work. The main aim of a proposed solution in the long term is more effective implementation of services through a better understanding of the current needs of customers and predictive analysis of their future needs. This will enable the urban public transport organizer to adapt its offer to the needs of current and potential passengers.

### 5.2. Implementation Plan

Thanks to the determination of specific application components, it will be possible to eventually combine individual modules and create an automatic data flow, which will generate savings in the future. Savings will result from manual work reduction, which a company has to perform or subcontract.

The implementation of described application consists of steps specified in Table 2. It is worth emphasizing that creating an application consisting of modules, the correct definition of assumptions, and the implementation of all modules determines the effectiveness of the whole application. Therefore, changing the sequence of actions or skipping some of them is not recommended [42].

### 5.3. Initial Configuration of Power Automate

The start of the project initiates the creation of the main account in a cloud service (Microsoft Office 365), which allows access to all the tools needed for further work. Once an account is active, the next step is to initially configure Power Automate as a major part of an entire feedback analysis process.

It is necessary to use the 'Automated cloud flow' option during the configuration. It will define the creation of a data flow. Then, appropriate 'triggers' (i.e., elements that will be components of the entire application) must be selected. They can be defined as situations that cause repetitive action in the automation process [43]. The applications that must be

chosen from the list of available triggers to create an initial data flow process are as follows: Microsoft Forms, Microsoft Office 365, Microsoft Excel Online, and Microsoft Outlook [44].

**Table 2.** Summary of the project implementation stages.

| No. | Stage | Description |
|---|---|---|
| 1. | Initial configuration of Power Automate | Account creation and initial configuration of space in Power Automate. |
| 2. | Configuration of Microsoft SharePoint | Creation of a new page in Microsoft SharePoint, including the structure of folders and subfolders for storing files. |
| 3. | Creation of survey in Microsoft Forms | Preparation of survey questions, the definition of an answer format and creation the visual side of a template. |
| 4. | Export of data collected in the form | Export of responses collected in the form to a separate registration file in Excel Online. |
| 5. | Creation of report in Power BI | Creation of a report in Power BI Desktop. A report needs to be shared in a workspace on a dedicated website. |
| 6. | Configuration of e-mail alert | Definition of the content of a message, which will automatically inform about new replies added to the form. |
| 7. | Process automation using Power Automate | Add all previous applications and tools to the previously created 'Power Flow' and configuration of connections. |
| 8. | Tests | Tests in Power Automate and manual tests consist in creating sample responses in the form. |

*5.4. Configuration of Microsoft SharePoint*

Microsoft SharePoint is a space that should be created at the beginning to structure space for data, files, and folders used in further work. Most often, in a company, there is already a pre-defined space in Microsoft SharePoint. In such a case, it should be appropriately configured to suit the assumptions of a designed tool [45].

The following steps of configuring a previously defined directory structure in Microsoft SharePoint are described below. The first step involves making the site's accessibility public, which allows all members of an organization to access a site. After clicking "Finish", the website should be created within a few minutes. Then, there is a need to create a new folder. An entire structure of folders and subfolders should be configured by repeating the abovementioned steps. After configuring a targeted hierarchy, it is possible to proceed to the following steps, which will be based mainly on Microsoft SharePoint in the context of data storage.

*5.5. Creation of Survey in Microsoft Forms*

Considering the feedback analysis process, it is good to remember that analysis cannot be started before gathering information from the client. Therefore, this stage is essential from the perspective of a company's service recipient. It is the only application element that has direct contact with a customer. This is critical information because a created survey should be user-friendly, considering good practices of creating forms. In addition, it should show a client that he is a crucial element of a business.

Firstly, there is a need to define questions to be included in the form. This is very important as they will perform several functions simultaneously. They must serve both as an entirely understandable method of assessing the company's activities in the eyes of a client and as a valuable source of information used to analyse collected data at a further stage of creating a report in the Power BI service. After adding the form title with a short subtitle and a company logo, it is worth configuring the entire form to match the visual identification of an organization (e.g., watermark or colour palette) [46]. Configuration of

the rest of the form and adding of questions is possible by the 'Add new' button. Then, it is necessary to enter the previously prepared questions and save a survey.

*5.6. Export of Data Collected in the Form*

Implementation of further activities is possible only if the first opinions are added. At this stage, the only repository of added questions is Microsoft Forms. Ultimately, each response added to the form generates a new row in a Microsoft Excel spreadsheet, which serves as a database. Firstly, a survey created in Microsoft Forms should be launched. Then, after clicking the 'Responses' button at the top of the page, simple statistics of answers given so far should be displayed. The 'Open in Excel' button enables start downloading the file in.xlsx format. Then, there is a need to run the file and send it to a previously defined location on Microsoft SharePoint. Through described actions, after the final configuration of an entire process in Power Automate, the file will intercept all responses provided by the previously prepared form.

*5.7. Creation of Report in Power BI*

Collected responses allow us to run the process related to analysing customer data and then draw conclusions based on them in Power BI. It is a crucial element of a described application because it enables full customization of a created report and individual visualizations thanks to advanced BI technology. It also gives a possibility of earlier data transformation, as well as an opportunity of creating additional measures and calculated values or implementation of machine learning.

Afterwards, data should be imported using a file located on Microsoft SharePoint. By importing data to a report, it will be possible to create visualizations and appropriate measures in a report (front-end layer). The report will present information updated in real-time in the following project stages. Further reporting work focuses on creating practical and useful measures and key performance indicators based on which a company will be able to change its offer based on collected opinions. Indicators are created in DAX (Data Analytical Expressions), which is implemented in all versions of Power BI [47].

Key Performance Indicators (KPI) are measures that a company uses to understand how well people or business units (e.g., departments, departments) manage with predefined strategic or operational goals (depending on the report audience) [48]. It is good to start by choosing a target theme for an entire report to create individual visualizations. It may be one of the built-in themes available in Power BI; however, to be consistent with a company's corporate identity, it is possible to customize them. The next step is to create visualizations that will be components of a report. Each visualization should have its unique description and application, thanks to which it will be handy for business purposes [49]. A report also consists of informally separated parts: title panel, KPI panel, filters panel, and visualization panel. A prepared final report is presented in Figure 2.

When a report is prepared correctly and fully useful, it can be published. Using a cloud service available in Power BI Web, a report can be available on any computer or mobile device with access to the Internet. This makes it possible to share a report via a link with anyone in a company, create roles with appropriate permissions and visibility, and manage to distribute a report in the network.

The most significant advantage of using Power BI Service is the ability to implement an automatic data refresh process. This is a crucial option in the context of the entire project because it allows data to be refreshed automatically at predetermined intervals. From that moment on, each time new information appears in the form, it will be automatically taken into account in subsequent report refresh sessions. Dates or specific times of data refreshing in a report can be adjusted to a particular company's needs, depending on the volume of data or frequency of its transmission.

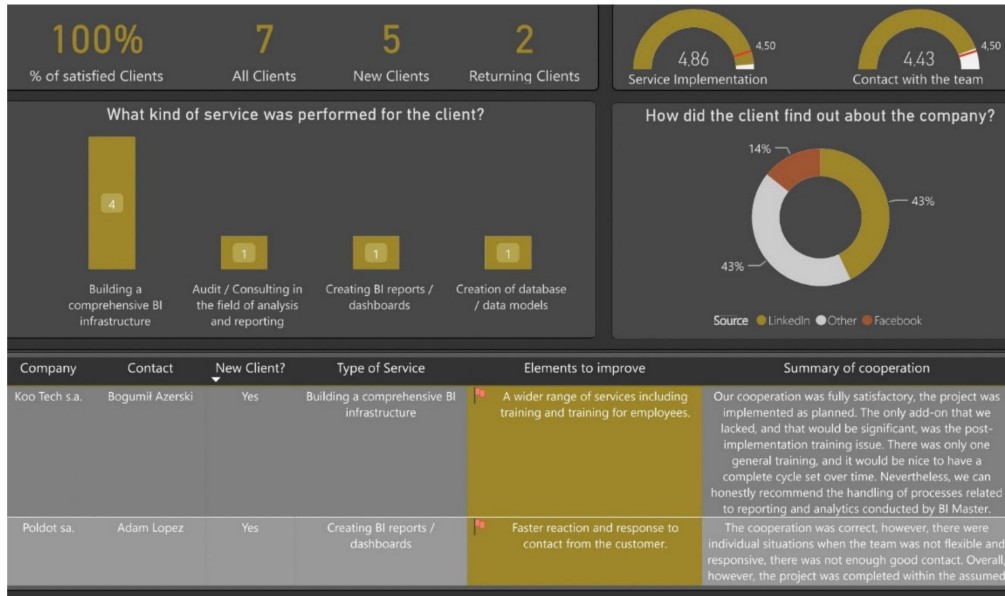

**Figure 2.** The final version of a sample report in Power BI.

### 5.8. Configuration of E-Mail Alert

In many business situations, an important factor influencing a company's immediate response to changes in its environment is a promptly received message with a notification that such communication has been sent. Most often, these are messages sent to group e-mail boxes or individual e-mail boxes of employees responsible for specific areas in a company. Due to information noise in most huge enterprises, the solution may be to automatically forward correspondence by e-mail directly to decision-makers. In such a system, a message is generated simultaneously as a customer gives an opinion by filling out a feedback form.

Configuring an e-mail alert is a task that requires restarting the Power Automate service as it is an activity entirely embedded in the automation logic implemented in Power Automate [50]. The content field contains complete information on the possibility of configuring individual data flows with links, which are helpful for a person responsible for managing an entire process. By a message structured in this way, a person dealing with a specific area can access all tools and components when a message appears in their e-mail inbox. An advantage of such a solution is the simplicity of accessing critical data in a process and the ability to edit individual steps of a process or documentation quickly.

### 5.9. Process Automation Using Power Automate

To create an expected automation level between applications, it is necessary to use Power Automate. A saved data flow should be run along with a configuration including mapping appropriate data from specific locations for each previously prepared application.

In the first step, defining a source of data (a form) is necessary. Data from a source will be downloaded each time when an answer is given. The selected form identifier should correspond to the name of the previously prepared survey. Then, the next step is a configuration of the task, which will be used to download data from the previously defined form.

It is essential to supplement information about a user with the required data, which is available thanks to the synchronization of Microsoft Office 365 accounts. To do it, expand the 'Download user profile (V2)' step, and then, in a field-defining user, a possibility of dynamic adjustment of an appropriate value should be used again. Adding this stage to an entire process makes it unnecessary to add the question to a form that collects information about the user. This would generate a risk of errors and a need to provide extra time to extend a form [51].

Afterwards, it is required to configure a data flow to a Microsoft Excel file placed on Microsoft SharePoint. In this step marked as 'Add a new row to the table,' two types of fields should be filled. The first one contains information about the source of the file, which should be selected. The second part of this step includes matching appropriate questions from a form to specific columns in a Microsoft Excel file. Then, it is necessary to configure in Microsoft Outlook an e-mail alert, which will be sent to people in a company responsible for a feedback analysis process. Alert will be sent each time a new answer appears in the form and will be passed to a report generated in Power BI [52].

*5.10. Tests*

The last stage of creating a tool includes tests carried out on several levels. The first level is internal testing, available as a built-in option in Power Automate. To start them, running a data flow in Power Automate is required, and then selecting the 'Flow controller' option will begin the process based on the Microsoft algorithms included in the application.

The second type of test is to send sample feedback via Microsoft Forms and manually check if responses have been properly sent to each application participating in an overall process. Firstly, it is required to validate the data available in a form and verify the correctness of an export process to a Microsoft Excel file, which stores data. Additionally, it is necessary to check that the data in the Power BI report has been refreshed correctly.

After performing all steps described above and conducting all required tests, it can be concluded that the tool is working correctly and is ready for operational implementation.

## 6. Case Study

An example of the developed tool is presented for the largest organizer of urban public transport in Poland, the Upper Silesian and Zaglebie Metropolis (Metropolis GZM). This institution is a metropolitan association established on 1 July 2017, based on the Act of 9 March 2017 on the metropolitan association in the Silesian Voivodeship [53]. It covers the area consisting of municipalities with solid functional links and a high level of socioeconomic development due to the industrial character of the region. The metropolitan area is inhabited by more than 2.4 million residents [54].

On 1 January 2018, the Metropolitan Authority established Metropolitan Transport Authority in Katowice (ZTM) to perform its statutory tasks. Therefore, ZTM formally realizes the tasks dedicated to the public transport organizer for 41 municipalities that make up the Metropolis GZM. Moreover, based on the appropriate agreements, ZTM also organizes public transport in 13 communes with solid functional connections with the metropolitan area. The entire area of ZTM operations is shown in Figure 3.

ZTM is an institution employing over 300 people, carrying out tasks in planning, management, and organization of public transport in an area of over 2.5 [thousand km$^2$]. From an organizational point of view, these tasks are performed by employees of various departments, i.e., transport, commercial, operational management, controlling, and administration.

The activities of ZTM aim to ensure a high level of service quality and passenger satisfaction, which led to an increase in the number of public transport users. As a result, it is possible to contribute to sustainable development goals [55]. Due to the specificity of the ZTM's core business, the number of employees, and the scale of operations (number of passengers, area of operations, number of transport means), ZTM deals with such research problems as selecting resources for tasks, vehicle routing problems, demand analysis or designing schedules. Various types of methods, such as genetic algorithms [56,57], ant algorithms [58], neural networks [59], or simulation tools [60], can be used to solve the described research problems, which are widely described in the literature. However, it should be noted that to apply individual optimization methods and examine their application's effectiveness, it is necessary to obtain information on the level of passenger satisfaction.

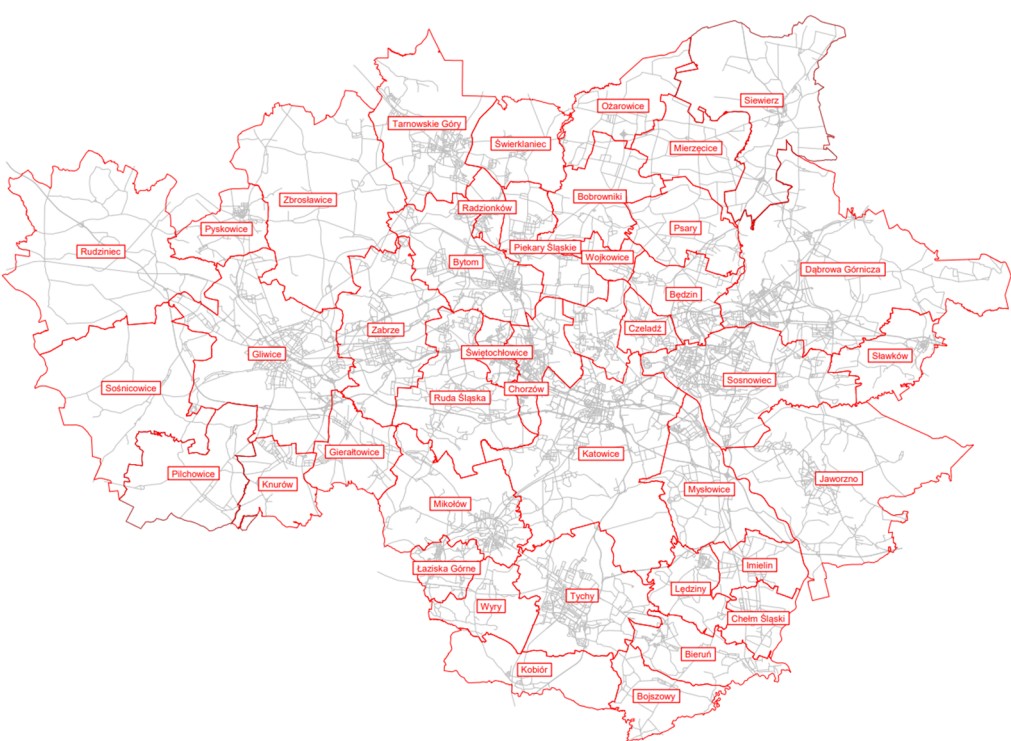

**Figure 3.** The area of the GZM Metropolis [61].

Collecting feedback in the urban public transport sector is a specific process connected with the following issues:

- The urban public transport organization requires constant improvement and considering passengers' needs, which is necessary to compete with individual transport.
- Grasping passengers' opinions is an endless and uninterrupted process.
- Public transport is characterized by big data, especially in urban agglomerations.
- Decision-makers responsible for the public transport organization expect data to be presented in an accessible and understandable way.

BI technology corresponds to the requirements as mentioned above. Thus, a tool based on BI technology is suitable for collecting feedback from public transport passengers.

An example of the application of the model presented in the article concerns the media monitoring process related to the information on the functioning of public transport. Information is one of the most important factors influencing the decisions made by residents and guests of the GZM Metropolis about the choice of transport means for everyday trips. Therefore, the verification of publications on the activities of the organizer of public transport (in this case, the Metropolitan Transport Authority) should be constantly carried out. The scope of the implemented model and the various dimensions in which the results can be compiled are shown in Figure 4.

The objective scope for the developed model, taking into account the activities of the Metropolitan Transport Authority in Katowice, includes:

- The number of publications broken down by individual media types (information providers to travellers); knowledge of the number of publications allows brand awareness to be built.
- Information reach is expressed in the number of potential contacts with the message assigned to a specific publication; it differs from the scope by introducing variables relating to the real behaviour of recipients, i.e., the ways and frequency of using the communication channels.
- Advertising Value Equivalent (AVE) is the valuation of a given message expressed in monetary units (in the case of ZTM, these are zlotys); it consists of estimating the value of a publication or broadcast of a given message based on the advertising price

lists for a given medium, the publication area, the number of page views, the number of unique users, or the duration of a program; it expresses the number of financial resources that would have to be spent if a given material were an advertisement; knowledge of the AVE makes it possible to improve the quality of media publications.

- Identification of the sources of the most desirable media titles; this information allows you to build an image advantage by working with the essential editorial offices on a given subject.

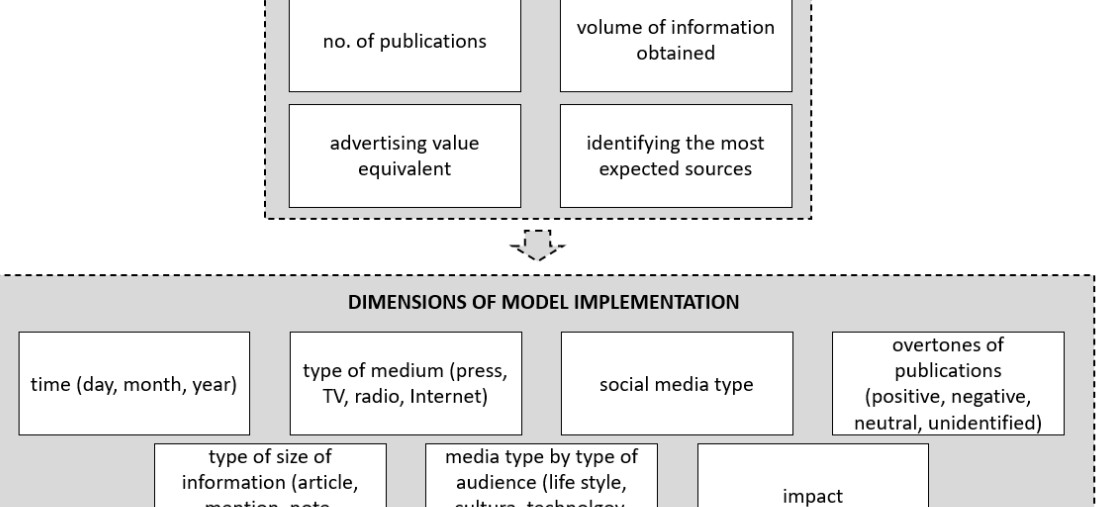

**Figure 4.** Scope of the implemented model and the various dimensions of the characterization of the results.

The results can be summarized in the following dimensions:

- Time, i.e., day, month, year.
- Type of medium transmitting the publication, i.e., tv, radio, press, social media.
- Type of social media, i.e., Facebook, Twitter, Polish radio, etc.
- Perception of a publication, i.e., the potential impact of the publication on shaping the image of the examined company in the media; the following types of perception were adopted:
  - Positive—when the message may have a positive impact on the media image of the company,
  - Negative—when the message is unfavourable to the image of the analysed company and may contribute to worsening its media image,
  - Neutral—when the message has neither a positive nor a negative impact on the brand image, or both the positive and negative information are contained in the publication.
- The size of the publication, i.e., the area occupied by information about the analysed company, taking into account the division into:
  - Article—the most extensive media publication; in the case of the press and the Internet, materials exceeding half an A4 page are defined, as well as those presented on one or more columns; for broadcast material, articles refer to information lasting at least 30 s.
  - News items—the medium-size media information; in the case of the press and the Internet, the publication does not exceed half a page of A4 format; in the case of radio and television information, it is material lasting at least 15 s.

○    Mention—a designation for the shortest materials, often one or several sentences long, which only mention the analysed company; in the case of radio and television news, these are materials that last less than 15 s.

- Type of media according to the kind of audience to which the content in these media is dedicated, i.e., lifestyle, general information, culture and entertainment, technology, automotive, finance, management, and others.
- Impact determining the strength of the influence of a given medium; it was assumed that the impact would be assessed based on a rating from 1 to 10, where the value of 1 is assigned to media reaching the narrowest audience and 10-to the media with the most significant scale of direct impact; the scale is common to all types of media, including social media, and refers to the reach indicator.

Bearing in mind the fact that every working day in the road and street network in the ZTM area, more than 1300 vehicles in bus, tram, and trolleybus transport are commissioned to carry out transport tasks, the provision of information on the operation of public transport has a significant impact on the quality of the entire transport system functioning. Therefore, the information must be accurate, efficient, and timely to the broadest possible audience. Therefore, the use of BI tools is beneficial in this case.

Table 3 and Figures 5–15 show selected media monitoring results obtained from using the BI tool for one year at the Metropolitan Transport Authority in Katowice.

**Table 3.** Selected general results from the one year.

| No. | Features of the Material Scope | Values |
|-----|-------------------------------|--------|
| 1. | Amount of information | 7 709 |
| 2. | Information reach | 2.1 billion |
| 3. | The volume of information obtained | 60 million |
| 4. | AVE | 11.2 million PLN |

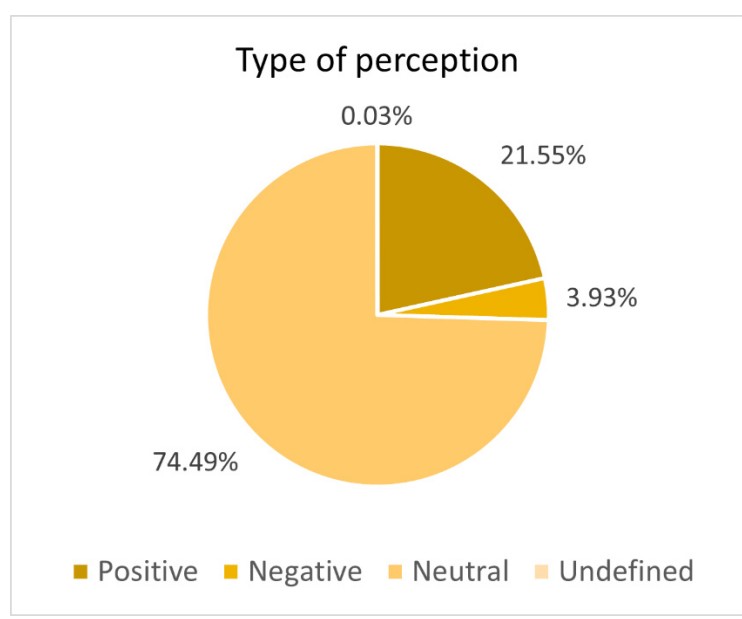

**Figure 5.** The general results of the media monitoring by type of perception.

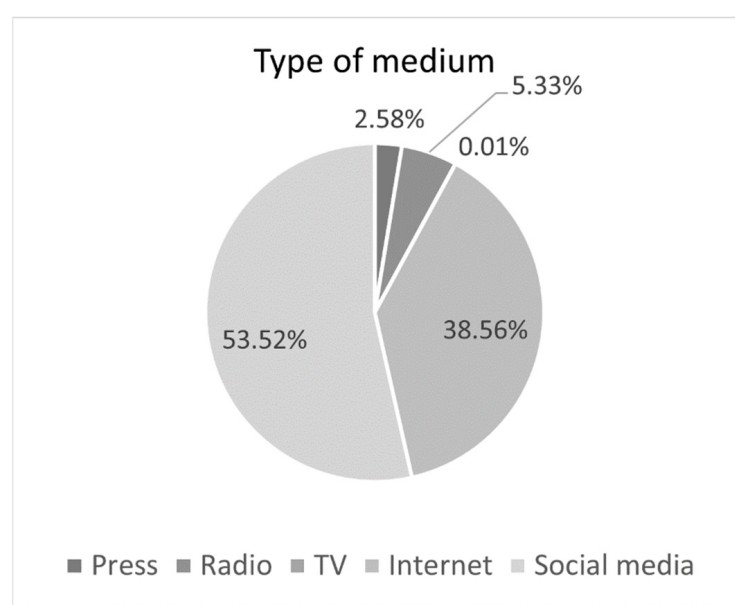

**Figure 6.** The general results of the media monitoring by type of medium.

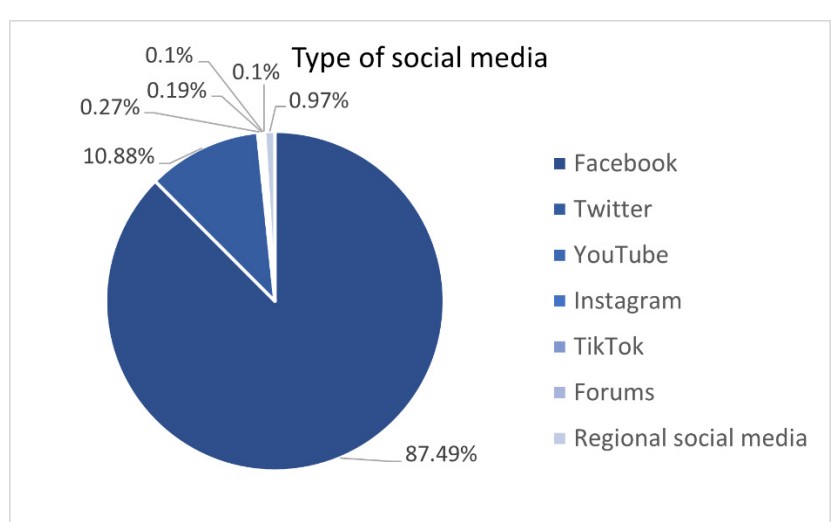

**Figure 7.** The general results of the media monitoring by type of social media.

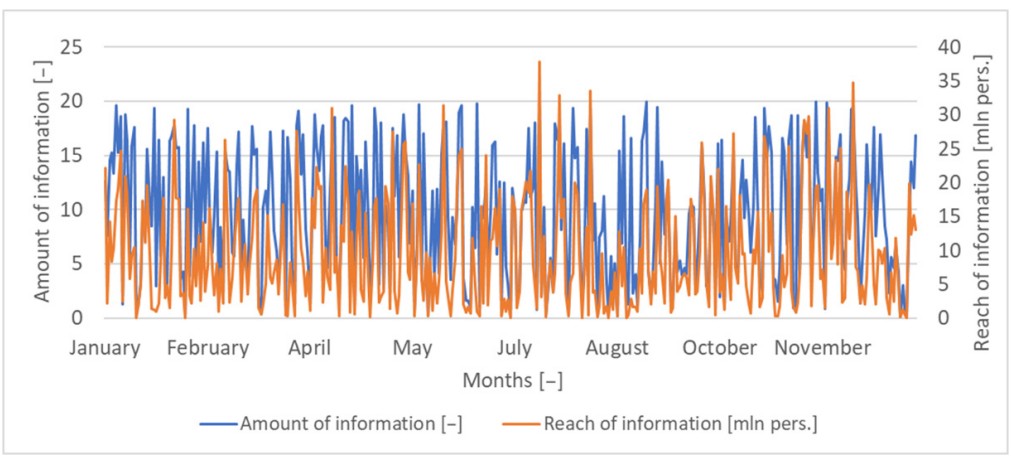

**Figure 8.** The distribution of the amount of information and reach of information on the Metropolitan Transport Authority.

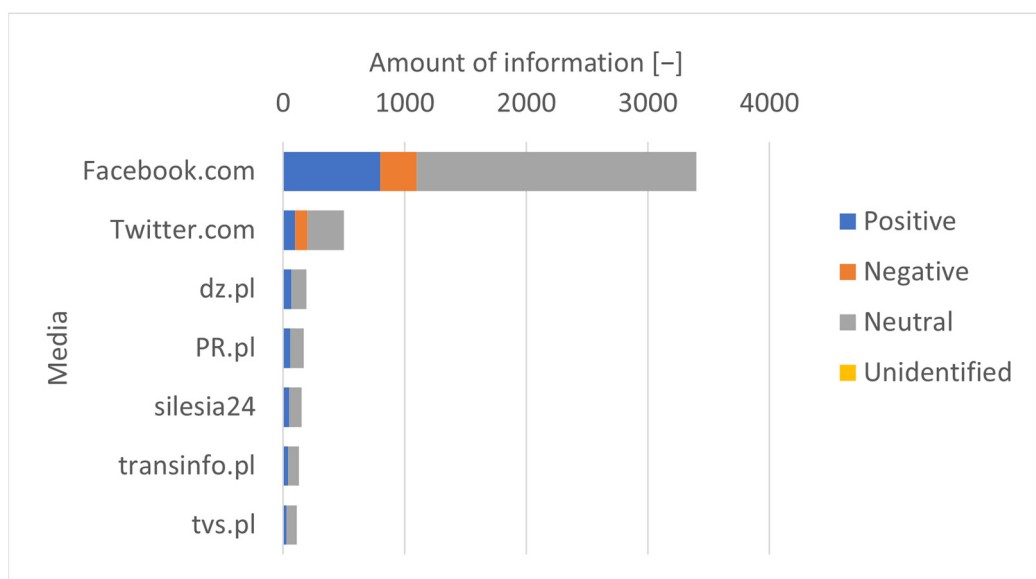

**Figure 9.** The media with the largest amount of information on the Metropolitan Transport Authority.

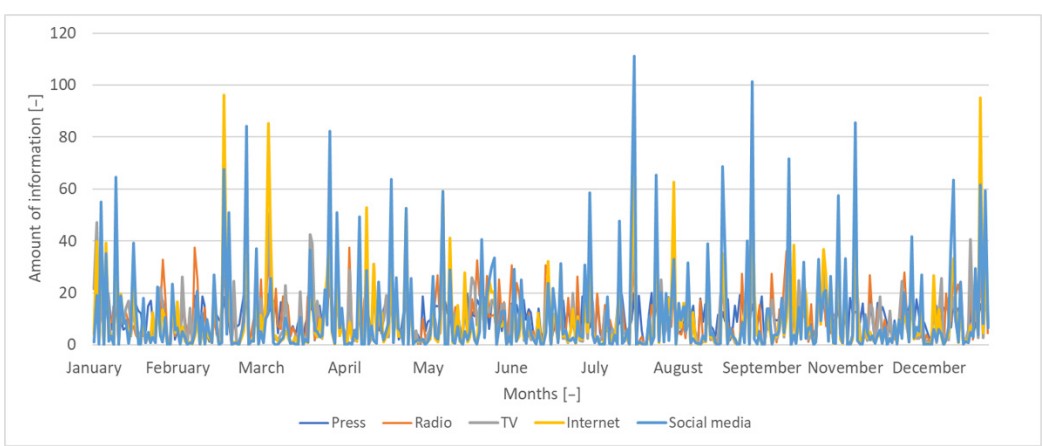

**Figure 10.** The distribution of the amount of information on the Metropolitan Transport Authority in various media.

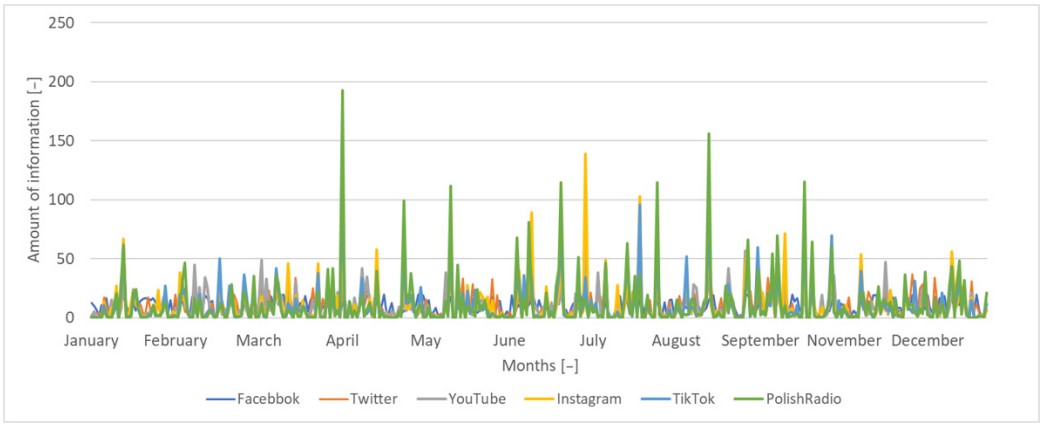

**Figure 11.** The distribution of the amount of information on the Metropolitan Transport Authority in various social media.

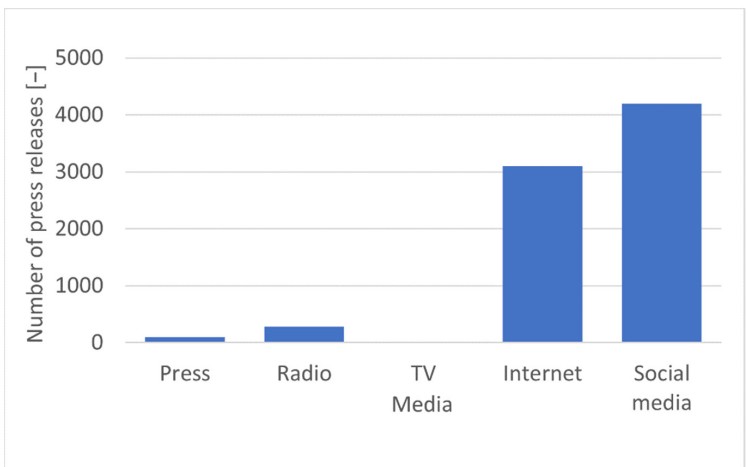

**Figure 12.** The number of press releases on the Metropolitan Transport Authority by medium and size.

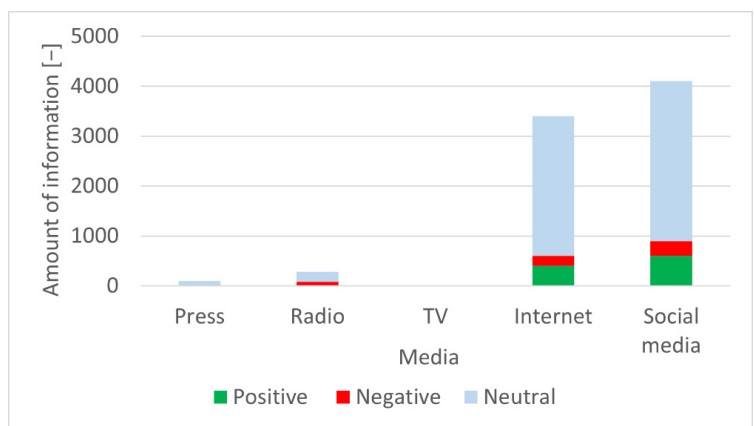

**Figure 13.** The amount of information on the Metropolitan Transport Authority by medium and type of perception.

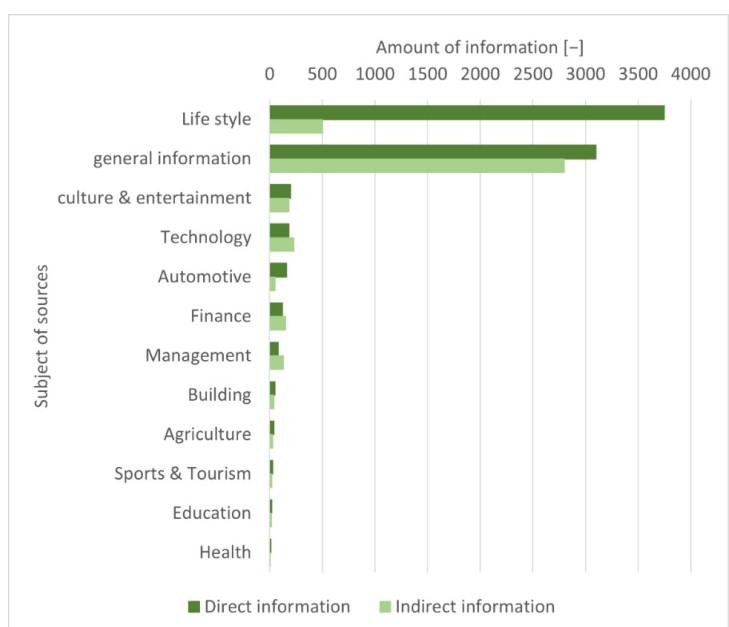

**Figure 14.** The amount of direct and indirect information on the Metropolitan Transport Authority by the subject of sources.

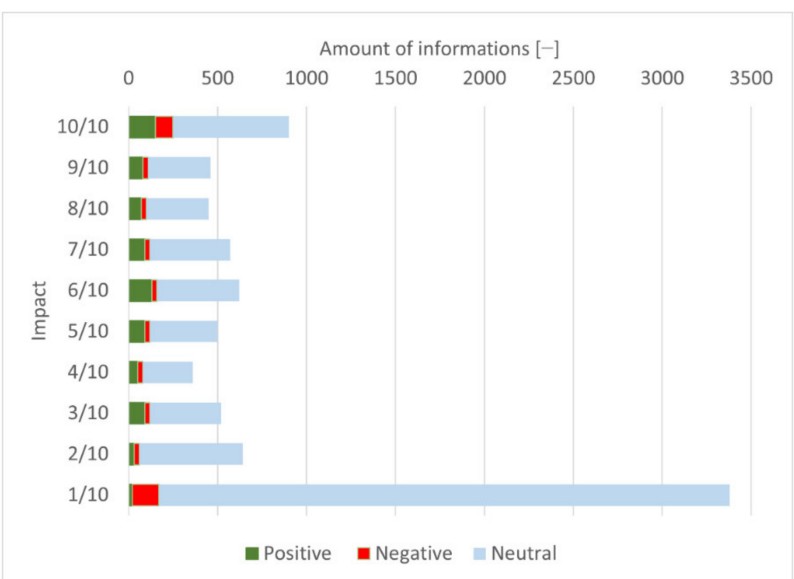

**Figure 15.** The amount of information on the Metropolitan Transport Authority by impact and type of perception.

Presented results of media monitoring include:

- The general results.
- The distribution of the amount of information and reach of information.
- The largest amount of information.
- The amount of information in the media.
- The amount of information on social media.
- The number of press releases by medium and size.
- The reach of information by medium and type of perception.
- The amount of direct and indirect information by the subject of sources.
- The amount of information by impact and type of perception.

The presented example shows the vast possibilities of using BI tools. Figure 5 illustrates the main results of the media monitoring, giving an overview of the media image of the Metropolitan Transport Authority. On this basis, it is possible to determine the perception of activities carried out by this entity (positive, negative, neutral, and undefined), the share of published information by media type (radio, press, TV, Internet, social media), as well as the percentage of published information in individual social media.

Figure 6 illustrates, in turn, the daily results on the amount of information provided and the reach of this information. There is a large variability over time due to the nature of the variable (i.e., information). Similar to Figure 8, with the difference that you can additionally see the division into the type of medium, and in Figure 9—into individual social media. In turn, Figure 7 clearly shows which social media enjoys the greatest amount of information provided and what is the perception of this information.

The characteristics presented in Figure 10 allow for the identification of the type of publication with the highest reach by media. However, it was impossible to define the type of publication on the Metropolitan Transport Authority. Figure 11 shows similar information as in the previous one, but with the additional possibility of determining the perception of given information.

By analysing the results presented in Figure 12, it is possible to determine the amount of direct and indirect information depending on the source. Moreover, it gives the possibility to assess how the information is received. The last figure shows the strength of the impact of a given medium.

Implementing one of the BI solutions, MS Power BI, in a company dealing with the organization of municipal public transport, gives a chance for a giant qualitative leap in

terms of understanding and, above all, appropriate use of data. Properly compiled data are sent to applications such as reports or managerial dashboards dedicated to employees of individual departments.

With BI-based tools, the end-user only sees the necessary information he needs to make the right decision. Figures and key performance indicators presented appropriately facilitate the employee's decision making in selecting the information distribution channel while searching for this data conventionally would take him a long time.

## 7. Discussion

Automating collecting feedback from clients using BI tools gives effective results, improving the quality of services provided. Classic methods have many limitations, including limited Big Data handling, a lack of possibility of automated data structuring, difficulties in data analysis, time-consuming data visualization, or susceptibility to human errors. The advantages of BI solutions are the ability to implement an automatic data acquisition, which enables analysis based on the current data.

The literature presents the research results related to the effectiveness of collecting feedback in various aspects. For example, the paper of Xinyan Z. et al. [62] contains an attempt to conceptualize and define a reliable measure of the social media impact during crises. The developed synthetic measure considers factors related to human behaviour. Analysis was performed using BI tools use. The research described in the article by Xinyan Z. et al. and Metropolitan Transport Authority case study presented in Section 6 are related to the different thematic areas. Still, in both cases, Twitter ranks among the highest in terms of the impact level on the studied area. In the case of the Metropolitan Transport Authority, Twitter ranks second, as shown in Figures 5–7 and 9.

Twitter information was also used as an analytical basis for risk management in the clothing industry, as presented in the article [63]. The role of big data analysis in social media in identifying and prioritizing supply chain risks is presented in this research. BI tools allow decision-makers to collect customer feedback and manage product distribution risk. The level of interest in the product was assessed based on, among other things, the number of tweets, likes, retweets, and comments. Similar analyses are also carried out by the Metropolitan Transport Authority for each media separately, as shown in Figures 5, 9, 13 and 15.

Moreover, Von Scheel H. et al. [64] present the possibilities of using Business Process Management (BPM) and Center of Excellence (CoE) by creating a customer-oriented process design. The client is the Internet user who provides access to feedback. The research results presented in this paper confirm the relevance of using BI tools.

Today, BI tools are a key added value in analysing the feelings related to the effects of the activities carried out. In the case of urban public transport, it is pretty important since the basic tasks of public transport organizations is to adapt the transport offer to the transport needs constantly. BI technology allows the level of satisfying the transport needs to be measured. However, it is worth underlining that BI should not be the only data source since new technologies and digitization does not yet cover the entire passenger population. This means that traditional communication channels such as e-mail correspondence, organizing meetings, videoconferences, telephone calls, or single-use forms should still be in-use to increase the availability and scope of feedback.

Considering the above, a concept of shaping the transport offer was developed based on feedback received from public transport passengers for the Metropolitan Transport Authority using BI tools. The diagram of the concept is shown in Figure 16.

Following the adopted concept, shaping an offer that meets the needs of passengers requires conducting research aimed at identifying important features related to passenger transport behaviour and then determining the groups of passengers to whom a specific offer will be addressed, tailored to their needs, requirements, and expectations. The concept assumes the development of the structure of feedback distribution channels based on the assessment of currently used ways of obtaining feedback and new methods of

collecting data. It is important to define such feedback distribution channels that will make it possible to effectively receive information on the quality of transport services provided by each passenger group. Different BI tools can be used for each feedback distribution channel. It allows for a quick process and aggregates the obtained research results to conduct comparative analyses. The duration of the feedback acquisition process may vary depending on the company's needs.

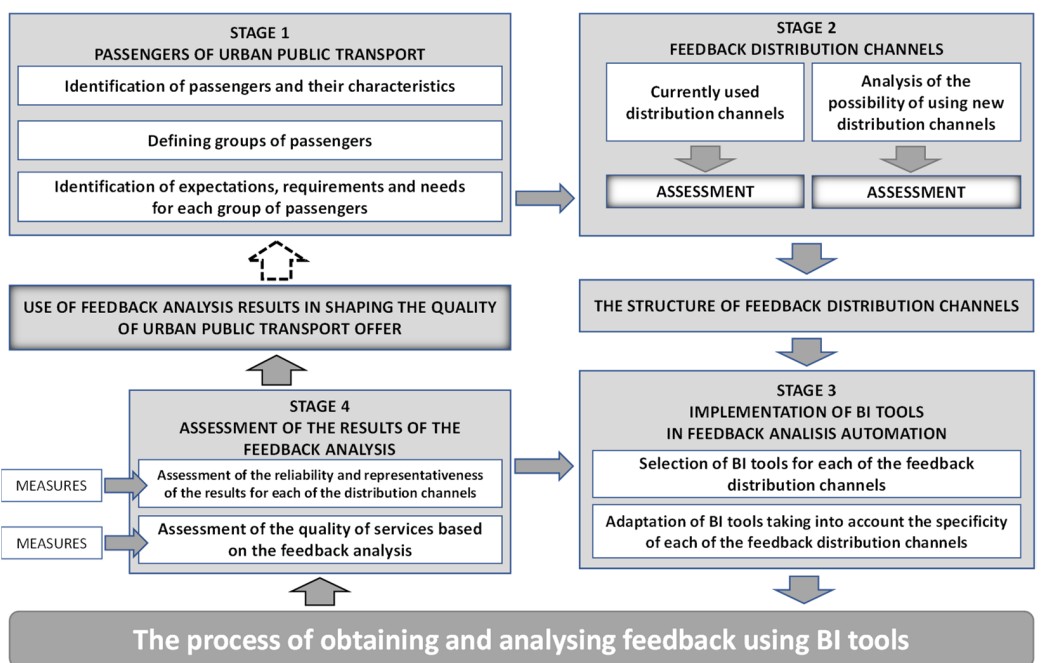

**Figure 16.** The diagram of the concept of shaping the urban public transport offer based on feedback using BI tools.

The fundamental element of the proposed concept is the evaluation of the results obtained based on feedback using BI tools. It is assumed that before assessing the quality of the provided transport services based on the obtained results, the results will be evaluated in terms of their reliability and representativeness. For this purpose, appropriate measures should be used.

It is worth emphasizing that the passenger's assessment of the transport services quality has to be constantly monitored. It is vital to adapt the offer to the expectations, requirements, and needs of various passenger groups, which may evolve and change due to multiple factors (e.g., social, economic, cultural, technological, etc.). This contributes to the necessity to keep up the transport offer with these progressive changes constantly. In the diagram shown in Figure 16, this link is marked with an arrow with dashed lines.

The proposed concept has a developmental character. It is assumed that the effects of the implementation will be presented in subsequent publications.

## 8. Conclusions

Companies dealing with the organization of urban public transport should constantly monitor passengers' reactions to changes in the transport offer. It enables making decisions concerning the shaping of the entire system of public transport. Due to the high complexity of this system, to draw interesting conclusions, it is necessary to use many different data sources. The structure of the data sources may turn out to be inconsistent.

In such situations, spreadsheets are often used, and the data are aggregated manually. Automating the entire data processing process is usually only possible to a minimal extent. Manual work is time-consuming and can generate errors and various types of inconsistencies. The analyses and reports generated in this way are often completely static and allow

you to look at only one precisely captured perspective. In addition, it should be noted that manual data collection often leads to human error, which can consequently generate extra costs (related, for instance, to the extra working time of an analyst). It should also be noted that some data sources do not allow direct data export. In the event of opinions sent by company employees, this necessitates obtaining specific accesses and authorizations from decision-makers.

Another solution, often considered by the companies, which could support the collection of passenger feedback is the use of ready-made reporting modules embedded into currently used systems (e.g., CRM). Unfortunately, the indicated tools have significantly limited functionality in many respects, including but not limited to:

- Compatibility with new, additional data sources.
- The scope and formats of the collected data.
- The sophistication of collected data analysis.
- Clear visualization of the collected information.

Additionally, the systems mentioned above with dedicated modules most often require the involvement of a third-party consultant on the part of the solution provider, who will be responsible for recording the submitted requirements and forwarding them to the development team for implementation. This generates further costs and creates an additional workload on the part of the company.

The solution to the problems mentioned above may be implementing the BI-based business application described in the paper. Thanks to the use of BI technology in the company, it is possible to:

- Accelerate data collection and analysis process significantly.
- Use of automatically refreshed and supplemented datasets.
- Obtain a consistent data structure.

The essence of BI is to rely on vast amounts of information that, when treated separately, does not present any practical value. On the other hand, when put together, using appropriate mechanisms and presentation techniques, it is an excellent material for analyses and predictions in the context of improving the quality of services offered by the company.

Despite their growing popularity in the context of performing advanced data analysis and visualization, the BI tools also have limitations. These limitations are mainly driven by the license type of the solution used. The functionality range is sometimes severely limited when using the free/trial version. Depending on the BI software provider, this may involve a smaller range of charts and visualizations for the application in the reports. It may also refer to a data volume limit at which the report will refresh. For free versions, some BI tools may also limit the use of predictive analytics or ML and AI modules. While this is not a problem in the initial development phase of an analytics platform, it is ultimately a limitation if you want to scale up your business and grow.

Using the example of the Power BI tool used in the paper, it can be pointed out that the software is unlikely to cope with transactional database systems with data volumes with hundreds of millions of records processed every day. Thus, the solution may not be a comprehensive tool to handle the entire process in terms of big data. Still, it will be ideal for tasks related to managing smaller process components such as feedback collection and analysis pointed out in the paper. Another limitation may be refreshing data capability. In free versions of Power BI, the system limits them to a frequency of a maximum of 8 times a day. This is a handicap in the context of data analysis in public transport, where information from, for instance, passengers, can feed the data repository at any time. Although this is not a significant obstacle for most analysts and people who work with BI tools daily, it is essential to mention the need for at least a minimum understanding/knowledge of programming to facilitate working with BI tools. In the case of Power BI, the user has at his disposal the DAX programming language, which is widely used for advanced calculation formulas. A lack of knowledge of this component results in the generation of only a basic report, which accordingly carries smaller business value for the company.

All the factors mentioned above allow accurate and prompt decisions to be made based on customer feedback. Based on the received information, a company can plan its further development, which will not entail additional costs related to an analysis of feedback because the presented tool allows service input from large numbers of customers. BI tools allow us to use not only the data but also the potential of the institution's employees because they have much more time to act based on data, which are additionally provided in a friendly, understandable, and error-free manner.

The article presents the implementation possibilities of the Microsoft BI tool supporting the automation of feedback analysis. An essential aspect of implementing Power BI is that it is most often a supplement to already owned systems and data and not a new system that replaces old IT tools. Therefore, it does not generate operational inconsistencies resulting from changes or interfere with the developed procedures. This enables a safe and cost-effective increase in the effectiveness of analyses.

The implementation of the proposed method can be carried out in any department of the company related to public transport organization, in particular: transport, commercial, operational management, controlling, and administration departments.

**Author Contributions:** Conceptualization, M.B., J.D., M.C. and J.M.; methodology, M.B. and J.D.; software, M.B. and J.D.; validation, M.B. and J.D.; formal analysis, M.B.; investigation, M.B., J.D., M.C., J.M., A.S. and R.Ż.; resources, J.D. and A.S.; data curation, J.D.; writing—original draft preparation, M.B., J.D., M.C., J.M., R.Ż. and A.S.; writing—review and editing, M.C., J.M. and R.Ż.; visualization, M.C., J.M. and A.S.; supervision, M.C., J.M. and R.Ż.; project administration, J.M.; funding acquisition, J.M. All authors have read and agreed to the published version of the manuscript.

**Funding:** This research was funded by the Warsaw University of Technology.

**Institutional Review Board Statement:** Not applicable.

**Informed Consent Statement:** Not applicable.

**Data Availability Statement:** Data are available in a publicly accessible repository.

**Acknowledgments:** The authors would like to gratefully acknowledge the reviewers that provided helpful comments and insightful suggestions on a draft of the manuscript.

**Conflicts of Interest:** The authors declare no conflict of interest. The funders had no role in the design of the study; in the collection, analyses, or interpretation of the data; in the writing of the manuscript; or in the decision to publish the results.

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
