# Peer review of "A Feedback Analysis Automation Using Business Intelligence Technology in Companies Organizing Urban Public Transport"

_sustainability, doi:10.3390/su141811740_

Round 1

Reviewer 1 Report

Congratulations on an interesting article done! However, I was concerned on these points, thus recommend a minor revision before publication, as follows:

1. The quality of Figure 1 should be improved. The same for Figure 2. The current quality of figures is too poor.

2. Section 7 should be considered to change to “managerial implications” or  discussions and conclusion”.

3. As this is an academic work, the English writing and linguistic quality of the paper should be improved with the help of an English native speaker. Also, I encourage the authors to use “passive voice” instead of “active voice” for the whole manuscript.

4. More discussion on the results. Comparative Analysis of methods should be added.

Reviewer 2 Report

This study presents a BI solution that can be an effective tool for automatically collecting passengers' opinions or experiences on the offered services.

1. Authors may enrich the introduction section by including more literature on research gaps and questions. 

2. Section 3. It is better to provide a comparison of different methods in tabular form highlighting merits and demerits of each method. 

3. Authors should highlight the reasons of using BI methodology in grasping passenger's opinion?  

4. Figure 5: the quality of figures need improvement for better readability. 

5. Section 6. there is no comparison of results with other methods or studies. it is required to present a comparison while discussing the results. 

6. What are the limitations of BI approach? discussion is required at the end of study findings. 

7. Section 7: it is better to name this section as conclusions

Reviewer 3 Report

The paper discusses automation of passenger feedback in public transit operations using off-the self software tools.  The demonstration of the proposed automation concept in a large public transport company in Poland is subsequently described.  It would be useful to present an evaluation of the impact on operations, first as a concept in this paper and perhaps as a formal evaluation in a subsequent paper.

Reviewer 4 Report

This paper can be published in the current version.

Round 2

Reviewer 2 Report

No comments